# The Patient, the Physician, or the Relationship: Who or What Is “Difficult”, Exactly? an Approach for Managing Conflicts between Patients and Physicians [note 1]

**DOI:** 10.3390/ijerph182312517

**Published:** 2021-11-27

**Authors:** Issam Tanoubi, Llian Cruz-Panesso, Pierre Drolet

**Affiliations:** Medical Simulation Centre, Centre d’Apprentissage des Attitudes et Habiletés Cliniques (CAAHC), Université de Montréal, Montréal, QC H2J 3T5, Canada; ilian.cruz.panesso@umontreal.ca (L.C.-P.); pierre.drolet@umontreal.ca (P.D.)

**Keywords:** communication, difficult patient, managing conflicts

## Abstract

It is the patient who consults, often at the last minute, the one you sigh over when you see his or her name on your list, the one who makes you feel powerless, and whom you would like to refer to a colleague. Every practicing physician has experienced being involved in a dialog of the deaf, with a patient refusing physicians’ recommendations, in a therapeutic dead end. Faced with such patients, the physician tries to convey scientific evidence to untangle the situation. When it does not work, he looks for other arguments, raises his voice, and avoids looking the patient in the eyes. When he is out of resources, trying to sound professional, he uses a sentence such as “I understand and respect your beliefs, but I am telling you what I learned in medical school!”. At the same time, his non-verbal behavior betrays more than a hint of irritation. Far from being caricatures, such situations generally result in the physician diagnosing or labeling the patient as “difficult.” This label is affixed on more than one patient in ten, and for all sorts of reasons. How, then, do you re-establish a relationship of trust? Or, even better, how do you avoid such labeling?

## 1. How Does a Patient Become “Difficult”?

Patients who show up with a “shopping list”, those who demand an immediate result, who are pessimistic, disrespectful, restless, or even malicious are all at risk of being labeled as “difficult” [1,2]. However, these encounters often reveal as much about the physician as they concern the patient.

We usually seek to see things according to our terms, rules, and frameworks of thought. Labeling a patient as “difficult” may allow us, as health professionals, to escape the fact that each individual has his values, priorities, reality, beliefs, and objectives, any one of which being more or less different from our own [3]. Should it be up to the patient to adopt the values and priorities of his physician? The patient identified as “difficult” by a physician remains an individual who is striving to optimize his quality of life. When consulting, the physician should not expect the patient to live up to the practitioner’s expectations instantly.

Whether applying the label “difficult” to a patient is proper or not, it certainly should not be considered an unwavering state of affairs. Most people, including healthcare workers, can become “difficult” under certain conditions or circumstances. Understanding, even welcoming the fact that a complex interaction with a patient is mainly contextual and reflects the stakes to be analyzed, makes it easier to address and influence such interaction adequately. It is incumbent on the practitioners to contemplate what could facilitate and reinforce the physician–patient dynamics in which they are involved and avoid actions that jeopardize it.

Removing the “difficult” label or never having to use it comes from realizing that one should adapt the approach to the patient’s situation. It cannot happen without a thorough understanding of the patient’s context, priorities, and short- and long-term objectives, leading to a carefully customized therapeutic plan.

Once the “difficult patient” label is affixed, it is not too late to fix it. Recent publications and scientific research suggest several helpful crisis management principles to withdraw the “difficult patient” tag. These tips are mainly based on our situational awareness and our consideration of the contextual understanding and the patient’s frame.

## 2. Principles of Crisis Management

Crisis management principles [4] (or crisis resource management) rely on the physician to display all the skills needed to manage a critical situation that puts the patient’s prognosis at stake [5]. These skills and principles have to be taught and need to be practiced, in significant part, through simulation training. The principles needed to address a “medical crisis” can also defuse conflicts and avoid entering a non-constructive physician–patient partnership to refuse care. Such situations can be considered medical crises that can compromise the quality of care and decrease compliance with the physician’s handling plan, thus jeopardizing patients’ well-being and prognosis.

### 2.1. Understanding the Patient’s Anger

Anger is an emotional reaction: uncovering its source is helpful to the physician–patient relationship. Anger usually stems from one’s perception of being offended, misled, or harmed. Anger, like frustration, is a reaction to the feeling that one has been dealt with inadequately or unfairly, whether justified or not. The perceived offender can be the “whole system”, a particular healthcare professional, a receptionist, or even the illness itself. Often, the patient is only seeking to be heard and recognized by the person caring for him. If the practitioner patiently lets the patient verbalize his anger and gently and empathically aims to find the cause of the frustration, it often subsides. The underlying causes can then be addressed in the same way a clinician addresses the cause of an illness. The “difficult” patient needs to feel listened to and accepted without judgment. This first part of the problem-solving approach offers empathy and allows patients to maintain a trusting relationship [6]. The Duke [7] team uses the mnemonic “NURSE” to label five types of continuer statements to enhance and structure the conflict-solving approach. These statements are emotion-centered: name (N) the patient’s emotion, understand (U) and legitimize the feeling, and respect (R), support (S), and explore (E) by asking the patient to elaborate on the emotion.

### 2.2. Understanding One’s Own Emotions and Counter-Transfer to Prevent a Conflict from Worsening

Frequently, only a few seconds are needed to exacerbate a conflict or prevent it from worsening and reaching a boiling point. Pouring fuel or water on a fire that has just started will result in a different outcome. Physicians often blame the lack of time as the main reason preventing them from adequately addressing a conflict. Still, several significant conflicts can be resolved or altogether avoided by investing only a modicum of time. When patients feel ignored or not listened to, when they feel that their needs are not considered, or even worse, they are being blamed, their anger simply increases. In such a situation, it is likely for the healthcare provider to become defensive, sometimes even overtly so, especially if his emotions are not under control. Telling patients that they are “inadequate” or “difficult” can only aggravate matters. People use verbal aggressiveness or insults to convey their despair and feeling of having exhausted every so-called “civilized” method to be heard and meet their needs. The blame inflicted on patients and the resulting shame that stems from it provides them with a justification to use any necessary means to protect their dignity and defend their cause in various ways [8]. The undoubtedly tricky task falls to the physician. He must ignore how the message is packaged by his patient and concentrate on the core of its content. Physicians, like most humans, naturally exhibit a predisposition for self-defense and fighting fire with fire when they feel cornered or blamed [9]. Resisting the urge to blame or react impulsively is critical to avoid an unnecessary, counterproductive situation.

### 2.3. Maintaining Neutrality and Showing Benevolence

Anger is an emotional reaction: uncovering its source is helpful to the physician–patient relationship The maintaining of neutrality and calm increases the chances of resolving a conflict. It improves our capacity to listen and to pay attention. Being neutral allows for an objective analysis of the situation and increases the chances of resolving the conflict without having our self-defense and self-protection mechanisms distracting us. The physician must set aside his interests and ego whenever there is a problematic interaction with a patient [8]. If the emotional storm has not passed, the practitioner must maintain a sympathetic ear: that is what will allow the storm to die down.

### 2.4. Identifying and Naming the Problem

For those of us who are physicians, identifying and trying to give a name to a patient’s problem proves that we are concerned for him and want to ally with him rather than become adversaries. Insults or a raised tone of voice are just clumsy attempts to express an inevitable frustration [10]. We have to make an effort to work WITH the patient and not AGAINST him. Recognizing this frustration and expressing empathy show the patient that we are trying to understand and place ourselves in the role of an ally instead of setting ourselves up as judges, even if we do not endorse his behavior at the moment. Recognizing and naming the patient’s frustrations, even while we calmly and convincingly justify our disagreement with them, can be a much more powerful tool than a display of “mechanical” empathy that is bound to be perceived as lacking sincerity.

### 2.5. Apologizing

The apology is a universal healing force for resolving interpersonal or intergroup conflict in which one party acknowledges responsibility for an offense. The request for an apology should occur following an event that may be humiliating for the patient. The latter will not stop brooding over this event and generating negative emotions.

“*It is as if the person who humiliated me is living rent-free in my mind. The offended parties become obsessed with the humiliating event, playing it over and over in their minds. They think about how they allowed it to happen, what they could have done differently, and whether and what kind of revenge to seek. Such ruminations can last for days, weeks, months, or years, and in some circumstances, a lifetime. These thoughts often interfere with work, other daily activities, and restful sleep. This preoccupation often leads to distraction from tasks at hand, poor judgment, and ill-considered behaviors*.”[11]

The excuse in this context offers therapy for the different feelings of humiliation. According to Lazare et al. [11], the therapeutic effect of apologies relates to two categories of feelings. The first includes sincerity, empathy, forgiveness, trust, dignity, and a sense of being considered. These items are not necessarily related to apologizing, and they can be transmitted subtly and non-verbally. They are, apart from apologizing, necessary for a positive interaction with the patient. The second category includes therapeutic mechanisms directly related to apologizing, such as verbalizing shared values, explaining, acknowledging the offense, or identifying the fault.

Sincere apologies are excellent tools to defuse a crisis and to win back a patient’s sympathy. “I am sorry that…” or “I am sorry for…” allows us to bring solace to the patient [12]. Some physicians are reticent to use such expressions, perhaps for fear of having them viewed as admissions of error and, as a result, causes for medico-legal reprisals [13]. However, as long as these apologies apply to the conflict per se and are not used in the context of a deviant practice (in the case of professional malpractice, other considerations come into play), they are without consequence for the physician and can be highly beneficial to resolve the situation. Even when accompanied by a connotation of culpability or when the problem is clearly out of our control, apologies are still powerful tools. The capacity of the practitioner to apologize is proof of his honesty and understanding of the patient’s frustration and can re-establish a feeling of trust [14,15].

### 2.6. Educating the Patient

When it comes to procedures, realities of the healthcare network, and specific responsibilities to the patient, his understanding is crucial to minimize possible areas of conflict with the physician [16]. Once again, the physician has to explain these responsibilities to the patient and train him accordingly. This education can be undertaken globally via the media, at the hospital level through local procedures (brochures, pamphlets), or at the individual level during a consultation. Explaining time constraints, asking the patient to prioritize his demands, following up on them to meet all expectations, and discussing why some needs may not be compatible with safety are examples of methods used to avoid a conflict related to unmet expectations. However, this technique, as it is physician-centered, might not be accepted by the patient. The physician should be able to refer the patient to other members of the healthcare team who are better suited to help answer all the patient’s inquiries.

It is also crucial to ascertain how well the patient understands his state of health (incomplete or erroneous information is frequently transmitted from online sources), discuss the evolution of the illness, and adjust the patient’s expectations regarding the success or failure of treatment [17]. The practitioner must seek and examine every piece of false information the patient may have acquired and attempt to correct it to avoid future misunderstandings [18]. In this respect, it is worth noting that Canadian patients’ level of health literacy can be disappointingly low. A total of 60% of adults and 88% of the elderly in Canada do not have the needed skills to find, understand, and use information to make the right decisions for their health [19]. People who demonstrate a low level of health literacy have a hard time understanding and using day-to-day health information [19]. All of these measures should help limit the number of patients who do not follow their treatment correctly, a figure as high as 40% in some specialties. The physician’s empathy towards his patient and the patient’s empathy towards the physician, acquired through awareness, can go a long way to improve communications and generate a better understanding of each other’s points of view.

## 3. The Communication Triad

Respecting the communication triad [20] is at the core of a productive physician–patient partnership. This triad includes attentive listening, verbal empathy (confirming or paraphrasing what the patient has said to show him we understand the situation), and maintaining eye contact on a physical level coupled with an open posture. Additionally, the physician must show his human side by introducing himself and welcoming the patient (for example, by shaking hands) while making sure he is physically and psychologically comfortable.

“*Empathy is more than listening, but the trainee who listens well is already some way down the road to showing empathy. Attentive listening is a prerequisite for identifying empathetic opportunities. Provide feedback on listening behaviors, especially appropriate eye contact. It is normal to look away while talking, but when listening eye contact is essential*.”[21]

The practitioner must also avoid complicated medical jargon and automated questionnaires. The quantity of information shared with the patient is also crucial. The physician must punctuate his information with encouragements and indications of sincere interest and personal commitment on his part. It must be supplied calmly, without any time constraints. The patient must feel that the length of time needed to address his condition adequately has been set aside by the physician. The practitioner’s part of the dialog must not contain any presuppositions regarding the patient and his questions or worries. Not assuming anything, asking questions, and displaying curiosity are often the best ways to avoid crises and to deal with or defuse them when they occur.

Effective communications between the physician and the patient improve the level of satisfaction of both. Satisfaction reduces potential complaints about malpractice and enhances the patient’s physical and mental health. Recognizing the importance of communications in the physician–patient relationship is crucial. It is at the heart of the physician’s competence and expertise. Holding actual power over the patient, the physician must accept that he is primarily responsible for finding the most effective way of communicating useful information, even when it is difficult to say or hear. Good communication also allows the physician and the patient to arrive at a joint decision regarding the most appropriate care while gaining commitment from the latter to take charge of his treatment [22].

Some healthcare centers have taken to illustrate, in the form of comic strips, situations or contexts which have led to a negative patient–physician interaction. These illustrations have improved the understanding of both parties’ points of view to create mutual and efficient empathy. These illustrations bring open questions that allow the patient to ponder his interactions with the physician while permitting the latter to understand better how the patient finds himself [23]. Questions such as “What does the physician need to know about the patient?”, “What does the patient need to know about the physician?”, “What actions could the physician have performed or changed to improve his interactions with the patient?”, or even “What other actions could the patient have taken to improve his interaction with the physician?” help create and foster a reciprocal empathy [24].

## 4. A “Difficult” Patient or a “Difficult” Physician?

Respecting current research generally deals with the characteristics of the “difficult” patient [25]. Rare are those that document physicians’ character traits and practices who express frustration with some of their patients. It seems that younger physicians experiencing some extra-professional stress or lacking communication skills are more likely to report having “difficult” patients. More seasoned physicians are less likely to label patients as “difficult”, probably because they are more experienced and possess greater mental flexibility than their young counterparts. It is expected that experience can significantly influence the dynamics of the physician–patient relationship and its difficulties, whether it is to amplify or limit them [26].

It has also been observed that physicians report having more “difficult” patients presenting with psychological rather than physical symptoms. Even if most physicians admit the extreme importance of a patient’s psychological problems, it would seem that certain practitioners are better prepared to solve a biomedical than a psychological ailment. The physician’s belief in the psychosocial scope of his practice also appears to be a determining factor. Family physicians are among the physicians who report the least number of “difficult” patients compared to medical specialists. Other studies concentrate on the harmful effects that a high number of working hours can have on a physician’s well-being, his partnership with the patient, and, probably, on the level of tolerance and mental flexibility needed to manage a crisis with a patient [27].

## 5. Conclusions

The fact that a physician has labeled a patient as “difficult” should not impact dealing with him. Instead, this label should invite the physician to respect the significant rules of communication, keep behaving normally, and adapt to the patient’s needs [23]. Such an approach facilitates the relations with “difficult” patients and prevents sticking such a label on many others. Managing the “difficult” patient is an expertise in itself, and teaching it should be part of the curriculum for medical students, residents, and all practicing physicians. It can be taught effectively via simulation and workshops by putting actors into situations [21]. The skills needed for preventing a patient from becoming difficult can also be acquired with the help of partnering patients dedicated to the education and evaluation of future practitioners [28]. Future research would be interesting to examine the impact of learning the management of the “difficult patient” on the communications difficulties encountered by the healthcare professionals. This education can be offered to medical students and integrated into the medical curriculum or health professionals as part of continuing professional development. The current statistics of the prevalence of this situation are well known, and it would be interesting to compare them following an educational intervention in different populations of health professionals.

## Data Availability

Not applicable.

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
