# Peer review of "The Patient, the Physician, or the Relationship: Who or What Is “Difficult”, Exactly? an Approach for Managing Conflicts between Patients and Physicians†"

_ijerph, 2021, doi:10.3390/ijerph182312517_

Round 1

Reviewer 1 Report

Since the selected format of the publication is an essay, the scientific rigor in terms of literature analysis and research methodology is not required.

However, the following adjustments are highly recommended:

  1. to remove "grounded theory approach" from the title as irrelevant in the light of the content of the essay.
  2. additionally to formulate soft of the conclusions, or further idea for research or any concrete take-away for the reader. the current last section could be restructured in two parts - discussion and conclusions

Author Response

  1. to remove "grounded theory approach" from the title as irrelevant in the light of the content of the essay.

Thank you for the comment. We edited the title to "The Patient, the Physician, or the Relationship,
Who or What is" Difficult "Exactly? An Approach for Managing Conflicts Between Patients and Physicians".

2. additionally to formulate soft of the conclusions, or further idea for research or any concrete take-away for the reader. the current last section could be restructured in two parts - discussion and conclusions

Thank you for your comment. Indeed, we inadvertently omitted the subtitle "conclusion" in the submitted version of the manuscript, and we added it in the appropriate place. The conclusion
now includes a take-home message about the definition of "difficult patient," a key message about the management of the "difficult patient" situation, the importance of integrating the
management principles of the "difficult patient" in the educational curriculum of medical students
and finally a possible research track on the manuscript topic.

Reviewer 2 Report

Dear Authors,

This essays is interesting topic to discuss and publish.

My suggestion are following:

  1. The sub-sections in part of Principles of communication management are revised needed (number isn't sequentially)
  2. Set unbold for "1. Understanding the patient's anger"
  3. Margins and spacing in paragraph must following the template

Please add some data, like deep interview with physicians and patients. describe with time, hospital location, patients diagnose, (demography) as an evidence in this essay to strengthen the validity of your arguments.

Author Response

1. The sub-sections in part of Principles of communication management are revisedneeded (number isn't sequentially)

  1. Thank you. Corrected.

2. Set unbold for "1. Understanding the patient's anger"
Thank you. Corrected.

3. Margins and spacing in paragraph must following the template
Thank you. Corrected.

4. Please add some data, like deep interview with physicians and patients. describe with time, hospital location, patients diagnose, (demography) as evidence in this essay to strengthen the validity of your arguments.

Thank you for this very interesting suggestion. We have added two paragraphs with references within the text, mainly to support the importance of apologies and empathy when communicating with difficult patients.

Reviewer 3 Report

  1. Please clarify your use of "deaf" in lines 19-20. You are not referring to those who cannot hear, but I think you are referring to patients who do not accept your recommendations.
  2. Link the principles of crisis management into your introduction and the other for a smooth transition in all sections. 
  3. The title states this is a grounded theory approach. Grounded theory is a research methodology. Terminology is incorrect. 

Author Response

Please clarify your use of "deaf" in lines 19-20. You are not referring to those who
cannot hear, but I think you are referring to patients who do not accept your
recommendations.

Thanks for the suggestion. We have sharpened the meaning of the term "deaf" in the sentence.

2. Link the principles of crisis management into your introduction and the other for a smooth transition in all sections.

Thanks for your suggestion. We have added the following paragraph before discussing the crisis management principles.
« Once the “difficult patient” label is affixed, it is not too late to fix it. Recent publications and scientific research suggest several crisis management principles mainly based on our situational awareness and contextual understanding that led to the communication
issues. »

3. The title states this is a grounded theory approach. Grounded theory is a research methodology. Terminology is incorrect.

Thank you for the comment. We edited the title to "The Patient, the Physician, or the Relationship, Who or What is" Difficult "Exactly? An Approach for Managing Conflicts Between Patients and Physicians".

Reviewer 4 Report

This is an excellently written article about managing conflict between physicians and patients. The title describes the article as "a grounded theory approach." There doesn't appear to be any information in the article to substantiate the claim. Grounded theory is not defined or used as a framework for analysis. The article is not a report of research, but rather a literature review.

Author Response

The title describes the article as "a grounded theory approach." There doesn't appear to be any
information in the article to substantiate the claim. Grounded theory is not defined or used as
a framework for analysis. The article is not a report of research, but rather a literature review.

Thank you for the comment. We edited the title to "The Patient, the Physician, or the Relationship,
Who or What is" Difficult "Exactly? An Approach for Managing Conflicts Between Patients and
Physicians".

Round 2

Reviewer 3 Report

Sentence fragment on lines58-60.

Additional information and organization is much better. This is an informational article for those dealing with patients who exhibit difficult behavior. Please ensure all language is written as "person-first", i.e. non-biased. 

#6. Educating the patient is essential. However, lines 170-`173 suggest the physician tell the pt that there are time constraints to education, asking questions, etc. This tactic does not work,  is physician-centric, nor is it evidence-based. Patients do not care about our time. The physician should answer those questions that are specific to the medical intervention suggested, then refer the patient to another member of the team, and tell the patient that there are others involved in their care who are better suited to help answer questions: the RN, or a nurse practitioner, or pharmacist, or cancer navigator, or any others who should be part of the healthcare team.

Author Response

Response to reviewers

Revised Manuscript

ijerph-1410326

The Patient, the Physician, or the Relationship, Who or What is "Difficult" Exactly? An Approach for Managing Conflicts Between Patients and Physicians.

Issam Tanoubi

I would like to thank the reviewer for the thoughtful review and his crucial suggestion about the limits of explaining time constraints to the patient.

Hereby are the point-by-point responses to the comments:

Reviewer #3

  • Sentence fragment on lines58-60.

The sentence has been rewritten for better fluidity and understanding. Thank you for the suggestion.

  • Additional information and organization is much better. This is an informational article for those dealing with patients who exhibit difficult behavior. Please ensure all language is written as "person-first", i.e. non-biased. 

We have revised the manuscript to ensure the absence of bias. Thank for the suggestion.

  • #6. Educating the patient is essential. However, lines 170-`173 suggest the physician tell the pt that there are time constraints to education, asking questions, etc. This tactic does not work,  is physician-centric, nor is it evidence-based. Patients do not care about our time. The physician should answer those questions that are specific to the medical intervention suggested, then refer the patient to another member of the team, and tell the patient that there are others involved in their care who are better suited to help answer questions: the RN, or a nurse practitioner, or pharmacist, or cancer navigator, or any others who should be part of the healthcare team.

This is a significant remark. Thanks for bringing it up. The idea behind educating the patient about time constraints was setting their expectations and avoiding the frustration related to unmet expectations. It is true, as you have suggested that this can also lead to patient frustration. We, therefore, highlighted in our paragraph the importance of referring the patient to other members of the care team to answer his questions without time constraints.

"Explaining time constraints, asking the patient to prioritize his demands, following up on them to meet all expectations, and discussing why some needs may not be compatible with safety are examples of methods used to avoid a conflict related to unmet expectations. However, this technique, as it is physician-centered, might not be accepted by the patient. The physician should be able to refer the patient to other members of the healthcare team who are better suited to help answer all the patients' inquiries".